# Modeling of Wood Surface Ignition by Wildland Firebrands

**Oleg Matvienko** [1,2] **, Denis Kasymov** [2,*] **, Egor Loboda** [2] **, Anastasia Lutsenko** [2] **and Olga Daneyko** [1]

[1] Department of Physics, Chemistry and Theoretical Mechanics, Tomsk State University of Architecture and Building, 634003 Tomsk, Russia; matvolegv@mail.ru (O.M.); olya_dan@mail.ru (O.D.)

[2] Department of Physical and Computational Mechanics, Tomsk State University, 634050 Tomsk, Russia; loboda@mail.tsu.ru (E.L.); lu.anastasik@gmail.com (A.L.)

[*] Correspondence: kdp@mail.tsu.ru

**Abstract:** The probability of structural ignition is dependent both on physical properties of materials and the fire exposure conditions. In this study, the effect of firebrand characteristics (i.e., firebrand size, number of firebrands) on wood ignition behavior was considered. Mathematical modeling and laboratory experiment were conducted to better understand the conditions of wood ignition by a single or group of firebrands with different geometry. This model considers the heat exchange between the firebrands, wood layer and the gas phase, moisture evaporation in the firebrands and the diffusion gases of water vapor in the pyrolysis zone. In order to test and verify the model, a series of experiments to determine probability and conditions for ignition of wood-based materials (plywood, oriented strand board, chipboard) caused by wildland firebrands (pine twigs with a diameter of 6–8 mm and a length of 40 ± 2 mm) were conducted. The experiments investigated the firebrand impact on the wood layer under different parameters, such as firebrand size and quantity, wind speed, and type of wood. The results of experiments showed that the increase in wind speed leads to the increase in probability of wood ignition. Based on the received results, it can be concluded that the ignition curve of wood samples by firebrands is nonlinear and depends on the wind speed and firebrand size as well as their quantity. At the same time, there is no ignition of wood samples in the range of wind speed of 0–1 m/s. The ignition of wood is possible with a decrease in the distance between the firebrands with a decrease in the firebrand length. This result agrees more closely with the model.

**Keywords:** firebrands; WUI; wood; ignition; mathematical modeling; heat transfer; pyrolysis





## 1. Introduction

Wildland Urban Interface (WUI) wildfires are spread by forest fuels (FF) and construction combustible materials. The WUI wildfire problem is global [1]: many countries have faced large fires, which caused serious consequences (human, material, and economic losses) in recent years. In Canada, the WUI fire disaster at Fort McMurray in May 2016 destroyed over 2400 structures [2]. One can mark Camp Fire, which occurred in November of 2018 in the United States and which resulted in 85 lives lost and accrued an overall loss of approximately US $17 billion [3,4]. Wildfire season 2019–2020 in Australia burnt about 18.6 million hectares and destroyed over 5900 buildings. Such wildfires can occur in any type of combustible material, but commonly start with FF by natural (for example, lightning strikes) or artificial cause (for example, bonfires, uncontrollable prescribed wildfire, damaged or sparking power lines, arson). Essentially, the problem of wildfires in WUI is a problem of structural ignition, and the best approach aimed at reducing its severity is to reduce the potential of structural ignition [5–7].

One of the effects observed in large-scale wildfires is burning and smoldering firebrands formed in the fire front. They are capable to cover a distance of several kilometers and to initiate a new combustion center [8]. Articles [9–11] showed that the formation of smoldering firebrands is a process as a result of which FF, such as shrubs, trees, and

construction materials, are heated and separated into smaller smoldering firebrands during a wildfire. In case of WUI wildfires, the combustible fuel can be vegetation, such as trees or shrubs, as well as elements of construction materials. It is important to understand which materials tend to produce more or less firebrands in order to develop mitigation strategies in the case of catastrophic wildfires [12–14]. In addition, it is necessary to take into account the tendency of certain construction materials to ignite under the conditions of a point source exposure, which will improve the existing technologies of fire-prevention organization of construction in the case of such effect [15–18].

Recently, studies have been actively conducted aimed at the systematic study of the formation of firebrands, including both full-scale studies of the ignition of building structures [15,19–21] and modeling the interaction of firebrands with structures based on wood [16,22–28]. In particular, based on data of wildfires in cities [15] (Itoigawa, Niigata, Japan, 22 December 2016), as a result of which 147 structures were damaged, and 120 of 147 were destroyed, it became possible to get an idea of how firebrands are formed caused by WUI wildfires. This subsequently allowed one to develop an experimental proceeding for the ignition of full-scale roof structures, and the quantification of the degree of firebrands/embers formation during fire [29–31]. Several factors were found to be critically important to the ignition of fuels, namely, the ambient wind speed, the firebrand pile mass, and the density of the target fuel. For the configuration (a flat fuel surface from marine-grade plywood, oriented-strand board, or cedar shingles) considered in [28] ignition was found to be dependent on target fuel density; flaming ignition was additionally found to be dependent on wind speed. Higher wind speeds increased the rate of oxidation and led to higher temperatures and heat fluxes measured on the test surface. Manzello et al. [18] also looked at the effect of configuration using plywood and oriented strand board with ponderosa pine firebrands. The results show that there was a correlation between configuration, wind, and mass/number of firebrands.

There are a significant number of studies devoted to modeling the ignition of both inorganic and organic materials by metal brands and sparks [8,32–34]. Yin et al. [32] developed a theoretical correlation relating the time to ignition to the moisture content of a cellulosic fuel bed. Experimental data from firebrand ignition of a pine needle fuel bed were used to fit the correlation. Wang et al. [33] investigated the ignition propensity of hot, metal spheres on cellulosic fuel beds and developed a correlation relating the experimental data on the temperature for ignition to the moisture content of the fuel and the hot particle dimensions. These types of correlations are important to understanding experiments; however, they are impractical in describing the ignition process of many materials. In [34], a simple heat and mass transfer model was developed to describe the ignition process of cellulosic insulation materials due to firebrand deposition. However, the ignition process of common wood construction materials by organic firebrands is very different and studies like these are rare—only a few attempts have been made to simulate the ignition process of such materials by organic firebrands [35–37]. In [35], a simple engineering model to predict the ignition of common building materials by firebrand piles was developed. The model used time-varying heat transfer data and material properties developed by experimental testing. The model was used to predict ignition of select building materials with different firebrand piles but did not consider heat transfer between the wood and the gas phase, moisture evaporation in the wood sample, and the diffusion of water vapor in the pyrolysis zone.

One of the most complex and stochastic processes to understand in WUI fire spread is the ignition of recipient or "target" fuels by hot particles and/or firebrands [38]. Several models have been developed to calculate trajectories, combustion rates, and lifetimes of metal particles and burning embers lofted by the fire plume [39–41]. In order to model the transport of firebrands, the size and aerodynamic qualities of the firebrand, winds in both the thermal plume and wind field, and the firebrand burning characteristics must be taken into account.

The review [42] notes that one challenge necessary to model ignition from firebrands is a characterization of the transition between smoldering combustion to flaming combustion when fuels are ignited by glowing firebrands. Several numerical models of different accuracy degrees have been developed in recent years to simulate more accurately the ignition of a wildland fuel bed by a hot particle or firebrand. The motivation for their development was given by the "hot spot" ignition theory, which appears to provide a reasonable and simple approach for the prediction of particle size-temperature relationships for ignition [32,43–46].

To provide a more accurate insight into the spot fire ignition problem, Lautenberger and Fernandez-Pello [47] proposed a 2-D model to simulate the ignition of a powdered, cellulose, porous fuel bed by glowing pine embers. The model consisted of a gas phase coupled to a heat transfer and pyrolysis model that simulates condensed-phase phenomena. In his recent review, Fernandez-Pello [48] concluded that additional work is required to characterize practical materials and to better understand the boundary condition between the firebrand or heated particle and the fuel bed, as well as identifying the need of a 3-D model, to predict quantitatively the spot ignition problem.

There are many mathematical models of wildland fires, but only a few models take into account the contribution of burning and smoldering firebrands generated in the combustion zone, which are one of the main causes of fire spreading around the world. The theory has multiple limitations because it mostly does not take into account ongoing reactions in firebrands, the moisture content of fuels, lack of good conductive contact between fuels and firebrands, radiative feedback or external radiation and thermophysical properties of the fuel. At present, there is a crucial need to determine the parameters that affect firebrand burning conditions, as well as the further heat transfer from firebrands to combustible surfaces. It should be noted that the problem is not limited only to the practical aspects of fire protection of wood buildings and structures, but also provides a background for the physical and mathematical theory of wildland fires and can help in understanding the formation, transport of firebrands, and their potential to ignite fuels and produce spot fires. Understanding the generation, transport, and the ignition mechanisms of structural wood by firebrands is important for the prediction and prevention of wildfires propagation.

Although there have been separate studies on process of target fuel ignition by firebrands, most of the studies were qualitative and no quantitative probability models have been developed that relate ignition probability to various influencing factors. For a specific fuel, the probability of ignition depends on the firebrand characteristics, the environmental conditions, and thermophysical properties of the fuel. The focus of this research effort was to adapt and improve a 3-D physical-based mathematical model proposed in the articles [49–51] for modeling the ignition of a layer of wood by glowing wildland firebrand with its subsequent verification, experimental and numerical study the conditions of wood ignition by a firebrand. A characteristic feature of the model is that it considers heat exchange between the wood layer and the gas phase, moisture evaporation in the firebrands and the diffusion of water vapor in the pyrolysis zone. In addition, the model takes into account the effect of the group of firebrands on the ignition process.

## 2. Materials and Methods

### 2.1. Mathematical Model

Consider the ignition and combustion of a layer of wood with burning and smoldering firebrands. We will assume that the combustible components that make up the volatile pyrolysis products can be modeled with one effective combustible gas with the reactive properties of carbon monoxide. For a mathematical description, we introduce a Cartesian coordinate system: the OX and OY axes are directed horizontally. Let the OZ axis be directed vertically upward (Figure 1).

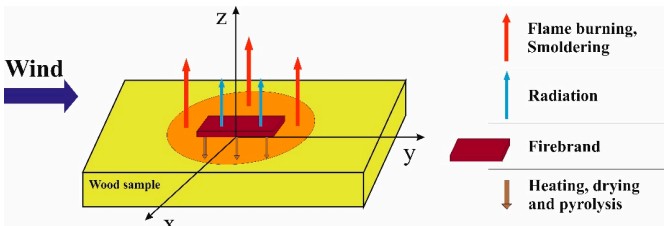

**Figure 1.** Process diagram.

The balance of thermal energy in a smoldering firebrand and a layer of wood is described by the thermal conductivity equation. This equation has the form for a smoldering firebrand [51]:

$$C_{fb}\frac{\partial \rho_{fb}T_{fb}}{\partial t} = \frac{\partial}{\partial x}\left[\lambda_f\frac{\partial T_{fb}}{\partial x}\right] + \frac{\partial}{\partial y}\left[\lambda_f\frac{\partial T_{fb}}{\partial y}\right] + \frac{\partial}{\partial z}\left[\lambda_f\frac{\partial T_{fb}}{\partial z}\right] + Q_{td}\phi_{td} + Q_{pyr}\phi_{pyr} + \varepsilon^*\frac{\left(U - \sigma T_{fb}^4\right)}{\delta}. \tag{1}$$

The thermal conductivity equation for a layer of wood is:

$$C_w\frac{\partial \rho_w T_w}{\partial t} = \frac{\partial}{\partial x}\left[\lambda_w\frac{\partial T_w}{\partial x}\right] + \frac{\partial}{\partial y}\left[\lambda_w\frac{\partial T_w}{\partial y}\right] + \frac{\partial}{\partial z}\left[\lambda_w\frac{\partial T_w}{\partial z}\right] + Q_{td}\phi_{td} - Q_{dr}\phi_{dr} + Q_{pyr}\phi_{pyr} + \varepsilon^*\frac{\left(U - \sigma T_w^4\right)}{\delta}, \tag{2}$$

where $C_f$, $\rho_f$, $\lambda_f$ are the heat capacity, density, and thermal conductivity of the firebrand ($f = fb$) and wood layer ($f = w$), $\delta$ is the characteristic firebrand size, which can be defined as the diameter of a ball with a volume equal to the volume of burning firebrands, $\varepsilon^*$ is the degree of blackness of the wood layer.

Radiant heat transfer is defined by the $P_1$ approximation. The radiation intensity is determined by solving the equation [51,52]

$$\frac{\partial^2 U}{\partial x^2} + \frac{\partial^2 U}{\partial y^2} + \frac{\partial^2 U}{\partial z^2} = 3\beta\chi\left(U - \sigma T_g^4\right), \tag{3}$$

where $\beta$ is the integral radiation absorption coefficient, $\chi$ is the integral radiation attenuation coefficient, $\sigma$ is the black body radiation constant.

The evaporation of moisture associated with the wood layer can be described as follows [53]:

$$\phi_{dr} = k_{dr}\rho_{fw}M_{hum}T^{-225}exp\left(-\frac{T_{dr}}{T_w}\right), \tag{4}$$

where $k_{dr} = 6\cdot10^5$ $\text{K}^{2.25}\cdot\text{s}^{-1}$, $T_{dr} = 6\cdot10^4$ K—pre-exponential factor and the activation temperature of evaporation. The reaction heat of vaporization was taken to be $Q_{dr} = 3\cdot10^6$ J/kg. $M_{hum}$—relative moisture content of wood sample.

The pyrolysis mass rate of the condensed phase (firebrand and layer of wood) can be calculated using the dependence [51–53]:

$$\phi_{pyr} = k_{pyr}\rho_f M_1 exp\left(-\frac{T_{pyr}}{T_f}\right), \tag{5}$$

where the thermokinetic parameters have the following values: $k_{pyr} = 3.63\cdot10^4$ $\text{s}^{-1}$, $T_{pyr} = 9.4\cdot10^3$ K, $Q_{pyr} = 10^7$ J/kg. $M_1$—mass fraction of dry organic matter.

The combustion rate of condensable pyrolysis products (coke) is described by the relation [51–53]:

$$\phi_{td} = k_{td}\rho_f M_{O_2}M_{pyr}exp\left(-\frac{T_{td}}{T_f}\right), \tag{6}$$

where $k_{td} = 10^6$ s$^{-1}$, $T_{td} = 10^4$ K—pre-exponential index and temperature of the heterophase combustion reaction activation. The reaction heat of the coke was taken to be $Q_{dr} = 1.2 \cdot 10^7$ J/kg. $M_{pyr}$—mass fraction of condensed pyrolysis products.

The change in the mass of the condensed phase (porous framework, water in a liquid-droplet state, and condensed pyrolysis products) is described by the balance equations:

$$\frac{\partial \rho_f M_1}{\partial t} = -\phi_{pyr}, \quad \frac{\partial \rho_f M_2}{\partial t} = -\phi_{dr}, \quad \frac{\partial \rho_f M_3}{\partial t} = \phi_{pyr} - \phi_{td}. \tag{7}$$

The equation of thermal conductivity of the gas phase can be written in the form [54–57]:

$$C_{p,g}\left(\frac{\partial \rho_g T_g}{\partial t} + \frac{\partial \rho_g v_x T_g}{\partial x} + \frac{\partial \rho_g v_y T_g}{\partial y} + \frac{\partial \rho_g v_z T_g}{\partial z}\right) = \frac{\partial}{\partial x}\left[\lambda_g \frac{\partial T_g}{\partial x}\right] + \frac{\partial}{\partial y}\left[\lambda_g \frac{\partial T_g}{\partial y}\right] +$$
$$\frac{\partial}{\partial z}\left[\lambda_g \frac{\partial T_g}{\partial z}\right] + Q_g \phi_g + \dot{h} + \beta\left(U - \sigma T_g^4\right). \tag{8}$$

The change in the enthalpy of the gaseous phase due to influx of gaseous pyrolysis products and water vapor produced during wood drying can be calculated as follows:

$$\dot{h} = \dot{h}_{pyr} + \dot{h}_{dr}, \quad \dot{h}_{pyr} = C_{p,\,CO}\left(T_f - T_g\right)\left(\phi_{pyr} - \phi_{td}\right), \quad \dot{h}_{dr} = C_{p,\,H_2O}\left(T_f - T_g\right)\phi_{dr}. \tag{9}$$

The reaction rate in the gas phase is described by the relation [58,59]:

$$\phi_g = k_g \rho_g^2 M_{O_2} M_{CO} \exp\left(-\frac{E_g}{R_g T_g}\right). \tag{10}$$

The reaction parameters are taken from [60,61]: $k_g = 2.28 \cdot 10^8$ m$^3 \cdot$kg$^{-1} \cdot$s$^{-1}$, $E_g = 104 \cdot 10^3$ J$\cdot$mol$^{-1}$.

To describe the processes of diffusion, mixing, chemical reaction, and combustion in the gas phase, in addition to the energy equation, the balance equations for the mass of the components were used $O_2$, $CO$, $CO_2$, $N_2$, $H_2O$ [62,63]:

$$\frac{\partial \rho_g M_{CO}}{\partial t} + \frac{\partial \rho_g v_x M_{CO}}{\partial x} + \frac{\partial \rho_g v_y M_{CO}}{\partial y} + \frac{\partial \rho_g v_z M_{CO}}{\partial z}$$
$$= \frac{\partial}{\partial x}\left[\rho_g \Gamma \frac{\partial M_{CO}}{\partial x}\right] + \frac{\partial}{\partial y}\left[\rho_g \Gamma \frac{\partial M_{CO}}{\partial y}\right] + \frac{\partial}{\partial z}\left[\rho_g \Gamma \frac{\partial M_{CO}}{\partial z}\right] - 2\frac{W_{CO}}{W_{O_2}}\phi_g + \phi_{pyr}, \tag{11}$$

$$\frac{\partial \rho_g M_{O_2}}{\partial t} + \frac{\partial \rho_g v_x M_{O_2}}{\partial x} + \frac{\partial \rho_g v_y M_{O_2}}{\partial y} + \frac{\partial \rho_g v_z M_{O_2}}{\partial z} = \frac{\partial}{\partial x}\left[\rho_g \Gamma \frac{\partial M_{O_2}}{\partial x}\right] + \frac{\partial}{\partial y}\left[\rho_g \Gamma \frac{\partial M_{O_2}}{\partial y}\right] + \frac{\partial}{\partial z}\left[\rho_g g \frac{\partial M_{O_2}}{\partial z}\right] - \phi_g, \tag{12}$$

$$\frac{\partial \rho_g M_{CO_2}}{\partial t} + \frac{\partial \rho_g v_x M_{CO_2}}{\partial x} + \frac{\partial \rho_g v_y M_{CO_2}}{\partial y} + \frac{\partial \rho_g v_z M_{CO_2}}{\partial z}$$
$$= \frac{\partial}{\partial x}\left[\rho_g \Gamma \frac{\partial M_{CO_2}}{\partial x}\right] + \frac{\partial}{\partial y}\left[\rho_g \Gamma \frac{\partial M_{CO_2}}{\partial y}\right] + \frac{\partial}{\partial z}\left[\rho_g \Gamma \frac{\partial M_{CO_2}}{\partial z}\right] + 2\frac{W_{CO_2}}{W_{O_2}}\phi_g, \tag{13}$$

$$\frac{\partial \rho_g M_{N_2}}{\partial t} + \frac{\partial \rho_g v_x M_{N_2}}{\partial x} + \frac{\partial \rho_g v_y M_{N_2}}{\partial y} + \frac{\partial \rho_g v_z M_{N_2}}{\partial z} = \frac{\partial}{\partial x}\left[\rho_g \Gamma \frac{\partial M_{N_2}}{\partial x}\right] + \frac{\partial}{\partial y}\left[\rho_g \Gamma \frac{\partial M_{N_2}}{\partial y}\right] + \frac{\partial}{\partial z}\left[\rho_g \Gamma \frac{\partial M_{N_2}}{\partial z}\right], \tag{14}$$

$$\frac{\partial \rho_g M_{H_2O}}{\partial t} + \frac{\partial \rho_g v_x M_{H_2O}}{\partial x} + \frac{\partial \rho_g v_y M_{H_2O}}{\partial y} + \frac{\partial \rho_g v_z M_{H_2O}}{\partial z}$$
$$= \frac{\partial}{\partial x}\left[\rho_g \Gamma \frac{\partial M_{H_2O}}{\partial x}\right] + \frac{\partial}{\partial y}\left[\rho_g \Gamma \frac{\partial M_{H_2O}}{\partial y}\right] + \frac{\partial}{\partial z}\left[\rho_g \Gamma \frac{\partial M_{H_2O}}{\partial z}\right] + \phi_{H_2O}. \tag{15}$$

The equation of state for the gas phase is:

$$\rho_g = \frac{p_g}{RT_g}\left(\frac{M_{CO}}{W_{CO}} + \frac{M_{O_2}}{W_{O_2}} + \frac{M_{CO_2}}{W_{CO_2}} + \frac{M_{N_2}}{W_{N_2}} + \frac{M_{H_2O}}{W_{H_2O}}\right)^{-1}. \tag{16}$$

The flow fields are described by the Navier–Stokes equations (Oberbeck–Boussinesq approximation), which in the Cartesian coordinate system have the following form [64,65]:

$$\frac{\partial \rho v_x}{\partial x} + \frac{\partial \rho v_y}{\partial y} + \frac{\partial \rho v_z}{\partial z} = 0, \tag{17}$$

$$\frac{\partial \rho v_x}{\partial t} + \frac{\partial \rho v_x^2}{\partial x} + \frac{\partial \rho v_y v_x}{\partial y} + \frac{\partial \rho v_z v_x}{\partial z} = -\frac{\partial p}{\partial x} + \frac{\partial}{\partial x}\left[\mu \frac{\partial v_x}{\partial x}\right] + \frac{\partial}{\partial y}\left[\mu \frac{\partial v_x}{\partial y}\right] + \frac{\partial}{\partial z}\left[\mu \frac{\partial v_x}{\partial z}\right], \tag{18}$$

$$\frac{\partial \rho v_y}{\partial t} + \frac{\partial \rho v_x v_y}{\partial x} + \frac{\partial \rho v_y^2}{\partial y} + \frac{\partial \rho v_z v_x}{\partial z} = -\frac{\partial p}{\partial y} + \frac{\partial}{\partial x}\left[\mu \frac{\partial v_y}{\partial x}\right] + \frac{\partial}{\partial y}\left[\mu \frac{\partial v_y}{\partial y}\right] + \frac{\partial}{\partial z}\left[\mu \frac{\partial v_y}{\partial z}\right], \tag{19}$$

$$\frac{\partial \rho v_z}{\partial t} + \frac{\partial \rho v_x v_z}{\partial x} + \frac{\partial \rho v_y v_z}{\partial y} + \frac{\partial \rho v_z^2}{\partial z} = -\frac{\partial p}{\partial z} + \frac{\partial}{\partial x}\left[\mu \frac{\partial v_z}{\partial x}\right] + \frac{\partial}{\partial y}\left[\mu \frac{\partial v_z}{\partial y}\right] + \frac{\partial}{\partial z}\left[\mu \frac{\partial v_z}{\partial z}\right] + (\rho - \rho_0)g. \tag{20}$$

The structure of the flow was modeled assuming that the air motion near the surface occurs in laminar mode, and the effect of pyrolysis products and evaporated water entering the gas phase on the flow structure is negligible.

The boundary conditions for the system of differential equations are determined by the type of boundary. For the windward side, homogeneous distributions of temperature, and also mass fraction of the gaseous phase components and the speed of the air are set.

$$x = -0.5 \text{ m}, \ z > 0: \ v_x = v\frac{z}{z_1}, \ v_y = 0, \ v_z = 0, \ T_g = 293 \text{ K}, \tag{21}$$

$$M_{O_2} = 0.76, \ M_{N_2} = 0.24, \ M_{CO_2} = 0, \ M_{CO} = 0, \ M_{H_2O} = 0, \tag{22}$$

$$x = -0.5 \text{ m}, \ z < 0: \ T_w = 293 \text{ K}, \tag{23}$$

here $v$—magnitude of speed at the height $z_1 = 1$ m.

Soft boundary conditions are set for the leeward, side, upper, and bottom boundaries:

$$x = 0.5 \text{ m}, \ z > 0: \ \frac{\partial v_x}{\partial x} = 0, \ v_y = 0, \ v_z = 0, \ \frac{\partial T_g}{\partial x} = 0 \text{ K}, \tag{24}$$

$$\frac{\partial M_{O_2}}{\partial x} = 0, \ \frac{\partial M_{N_2}}{\partial x} = 0, \ \frac{\partial M_{CO_2}}{\partial x} = 0, \ \frac{\partial M_{CO}}{\partial x} = 0, \ \frac{\partial M_{H_2O}}{\partial x} = 0, \tag{25}$$

$$x = 0.5 \text{ m}, \ z < 0: \ \frac{\partial T_w}{\partial x} = 0, \tag{26}$$

$$y = \pm 0.5 \text{ m}, \ z > 0: \ v_x = 0, \ \frac{\partial v_y}{\partial y} = 0, \ v_z = 0, \ \frac{\partial T_g}{\partial y} = 0 \text{ K}, \tag{27}$$

$$\frac{\partial M_{O2}}{\partial y} = 0, \ \frac{\partial M_{N_2}}{\partial y} = 0, \ \frac{\partial M_{CO_2}}{\partial y} = 0, \ \frac{\partial M_{CO}}{\partial y} = 0, \ \frac{\partial M_{H_2O}}{\partial y} = 0, \tag{28}$$

$$y = \pm 0.5 \text{ m}, \ z < 0: \ \frac{\partial T_w}{\partial y} = 0, \tag{29}$$

$$z = 0.25 \text{ m}: \ v_x = 0, \ v_y = 0, \ \frac{\partial v_z}{\partial z} = 0, \ \frac{\partial T_g}{\partial z} = 0, \tag{30}$$

$$\frac{\partial M_{O2}}{\partial z} = 0, \ \frac{\partial M_{N_2}}{\partial z} = 0, \ \frac{\partial M_{CO_2}}{\partial z} = 0, \ \frac{\partial M_{CO}}{\partial z} = 0, \ \frac{\partial M_{H_2O}}{\partial z} = 0, \tag{31}$$

$$z = -0.1 \text{ m}: \ \frac{\partial T_w}{\partial z} = 0 \text{ K}. \tag{32}$$

At the boundary between the condensed and gaseous phases, the flow speed components were assumed to be equal to zero due to the no-slip conditions: $v_x = 0$, $v_y = 0$,

$v_z = 0$. The conjugation conditions at the contact of firebrands, wood layer, and gas phase are set in modeling heat and mass transfer:

$$\lambda_g \frac{\partial T_g}{\partial n} = \lambda_w \frac{\partial T_w}{\partial n}, \ T_g = T_w, \ \lambda_g \frac{\partial T_g}{\partial n} = \lambda_f \frac{\partial T_f}{\partial n}, \ T_g = T_f, \ \lambda_w \frac{\partial T_g}{\partial n} = \lambda_w \frac{\partial T_f}{\partial n}, \quad (33)$$

$$\frac{\partial M_{O_2}}{\partial n} = 0, \ \frac{\partial M_{N_2}}{\partial n} = 0, \ \frac{\partial M_{CO_2}}{\partial n} = 0, \ \frac{\partial M_{CO}}{\partial n} = 0, \ \frac{\partial M_{H_2O}}{\partial n} = 0, \quad (34)$$

where $n$—is the direction of the normal to the surface.

At time $t = 0$, the temperatures of the wood layer and air were equal: $T_g = T_w = 293$ K. The firebrand temperature corresponded to the temperature of smoldering bark and branches of conifers $T_f = 850$ K [49].

### 2.2. Method of Solving the Problem

The above-presented equations form a closed complete system of equations that, at corresponding boundary conditions, can be used for determining the main characteristics of the heat transfer and ignition. Differential equations were solved numerically using the finite-volume method [66]. In accordance with this method, finite-difference equations are obtained by integration of differential equations over control volumes containing nodes of a finite-difference mesh. We have used the own code in this investigation. The mesh contained 2000 nodes in the x-direction, the 3000 nodes in the y-direction, and 1500 nodes in the z-direction, and it was bunched near the firebrands and in the regions with large temperature gradients. The SIMPLEC algorithm was used to satisfy the continuity equation [67]. It was believed that the convergence of iterations was achieved if the root-mean-square discrepancy for all the variables did not exceed 1%. The accuracy of the estimates made was verified in a series of calculations on a set of successively bunching meshes. The testing has shown that the decrease in the step of the base mesh along the axial and radial coordinates by two times changes the values of the main variables by no more than 1%.

Soft boundary conditions are set for the leeward, side, and upper boundaries. The wind speed is assumed to be constant and independent of height. At time t = 0, the temperatures of the wood layer and air were equal. The firebrand temperature corresponded to the temperature of smoldering bark and branches of conifers.

### 2.3. Experimantal Procedure

Studies of burning and smoldering firebrands generated in fires can be divided into three directions: generation of firebrands, transport of firebrands, and interaction of firebrands with fuel after landing. Many studies of firebrands have focused on the transport of firebrands, with little research on the generation of firebrands and ignition of fuels [68–70]. The development of science-based fire safety requirements for building materials and structures used in wildland–urban interfaces requires laboratory and semi-field experiments due to the difficulty and in most cases the impossibility of obtaining the necessary data during wildland fires.

Plywood, chipboard, and oriented strand board (OSB) are used as samples of wood construction materials, which are popular in building. The main parameters of the samples are presented in Table 1.

The laboratory setup was used to study the probability of ignition of wood building materials by burning and smoldering firebrands (Figure 2). The experimental procedure, as well as the main elements of the laboratory setup for dropping the firebrands are given in detail in [27,71].

**Table 1.** Parameters of the samples.

|  | **Plywood** | **Chipboard** | **OSB** |
|---|---|---|---|
| Type | broadleaf (birch) | coniferous | coniferous |
| Type of glue | urea | phenolic | phenolic |
| Size, m: | 0.100 × 0.100 | 0.100 × 0.100 | 0.100 × 0.100 |
| Thickness, m: | 0.021 | 0.018 | 0.018 |
| Density, kg/m³: | 705–725 | 700–720 | 570–590 |
| Moisture content, %: | 4.2 | 5.7 | 4.7 |
| Thermal conductivity, (Wm⁻¹ K⁻¹) | 0.14 | urea glue | urea glue |

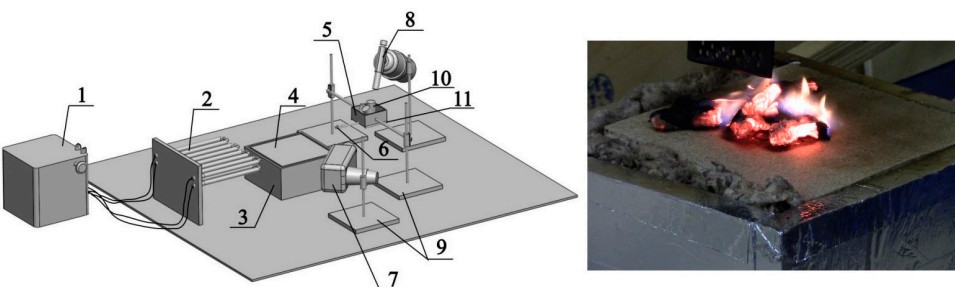

**Figure 2.** Schematic of the experimental setup: 1—-laboratory autotransformer; 2—-heating element; 3—-pallet; 4—-wood sample; 5—-cuvette; 6—-stopper; 7—-heat blower; 8—-burner; 9—-tripods; 10—-samples of firebrands; 11—-bracket.

The experimental setup included: a JADE J530SB infrared camera with an 2.5–2.7 micron optical filter that records the temperature in the range of 300–800 °C; a Canon HF R88 video camera for evaluating the ignition delay and the behavior of firebrands after falling onto the surface of the wood building material samples; an AND MX-50 humidity analyzer for controlling the humidity of the samples; an AND HL 100 laboratory scales for controlling the initial mass of firebrands and the mass of wood samples.

In all experiments, pine twigs with a diameter of 6–8 mm and a length of 40 ± 2 mm were used as the simulators of firebrands. Previously [22], the experiment was conducted on the ignition of wood samples from a pine construction board as a result of exposure to flaming and glowing pine bark firebrands. The experimental method was similar. It was concluded that the probability of ignition of wood samples increases with increasing firebrand size, as well as with increase in wind speed rate. The size of pine twigs (40 mm long) was chosen based on this, which is close in size to firebrands of pine bark, which has the highest incendiary potential for the chosen experimental parameters (30 × 30 mm and 5 mm thick).

The moisture content was determined using the AND MX-50 moisture analyzer (A&D Co., Ltd., Tokyo, Japan). The moisture content was 3.5 ± 0.2% for branches. Ambient temperature was 293 K during the experiments.

The samples were isolated from the environment with a heat-insulating material so that one of the surfaces remained exposed to heat from falling firebrands. A photo of the sample before the experiment is shown in Figure 3.

Glowing firebrands, which effects the surface of sample, are of particular interest in current research. The case when glowing firebrands that form during a wildfire can accumulate on the roof and in corners of buildings, fences, or find a way to get inside premises and ignite it is simulated in these experiments. The optimal firebrand ignition time was preliminarily selected (Table 2), at which the firebrand smoldering phase was achieved [22]. Recording the temperatures of the pine twigs was conducted using a thermal imaging camera JADE J530SB (CEDIP Infrared Systems, Croissy Beaubourg, France).

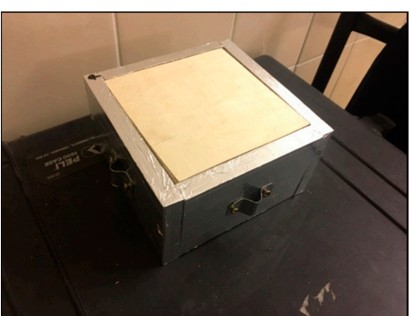

**Figure 3.** Photo of the sample before the experiment.

**Table 2.** Exposure time depended on firebrand size.

|  | Length, [mm] | Exposure time, [s] |
|---|---|---|
| Pine twigs | 20 | 15 |
|  | 40 | 20 |
|  | 60 | 25 |

In real fires, different wooden structures and fuel bed are exposed to firebrands and a number of natural factors, in particular, to a heated wind speed from the front of a fire. In the experiments, smoldering firebrands dropped on the wood samples were blown with a heat blower (Interskol FE-2000-E) by a heated air flow at a speed of 2.0 m/s and 2.5 m/s with corresponding temperatures of 60 °C, 110 °C. The flow speed was determined using a CFM Master 8901 anemometer with a measurement error of 2%. The choice of these speeds can be explained by the fact that with further increase in the speed, the firebrands were blown away beyond the area under study, and at lower speeds the probability of the sample ignition by smoldering and burning firebrands was very small. A nozzle was used to direct the air flow to the surface of the wood sample in the area of dropping of firebrands.

A series of experiments began with one firebrand, then two and up to 10 firebrands, thereby modeling the ignition of wood by both a single firebrand and "firebrand shower". Each experiment was repeated three times. If in one of the three cases the ignition occurred, the wood sample was considered to be ignited.

## 3. Results

### 3.1. Experiment

The analysis of the thermograms showed that the glowing of pine twigs is a uniform process compared with the glowing of pine bark. The averaged temperature maximum of glowing firebrands at the moment of land on wood layer was $1180 \pm 98$ K for pine twigs. The data are in good agreement with [72], where using laboratory experiment, the transfer of smoldering firebrands produced in forest wildfires by a heated gas flow was simulated.

In a series of the experiments, the probability of ignition of the preheated surface of plywood, PB, OSB, and spruce samples was evaluated depending on the type, size, and number of burning and smoldering firebrands interacting with this surface at different wind speeds.

Figure 4 shows the graphs of the probability of a surface of plywood, chipboard, and OSB samples depending on the size of glowing firebrands and their amount interacting with this surface at various wind speed.

Analysis of the graphs shows that as wind speed increases, the probability of wood ignition by firebrands of the same size increases. In particular, when the wind speed increases from 2 to 2.5 m/s, the minimum number of firebrands with a length L = 40 mm, sufficient to ignite wood, decreases from nine to four; eight to six; six to four firebrands for plywood, OSB and chipboard, respectively. The number of firebrands also affects the ignition of wood. The results coincide with the works of other authors [35]. The interaction between firebrand length and firebrand diameter (length times diameter) was also found

to be statistically significant. It should be noted that in accordance with the conditions of the experiment, the distance between the firebrands after falling on a wood layer, varied from 5 to 50 mm, depending on the firebrand size and their quantity.

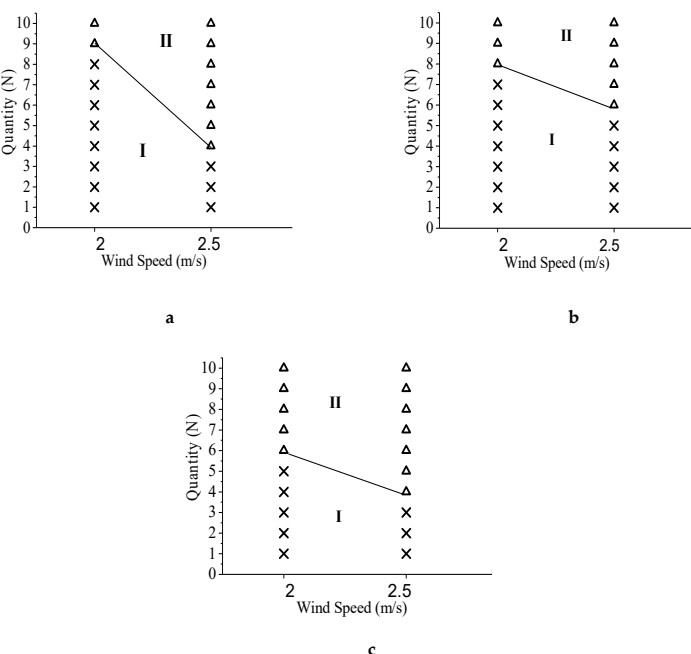

**Figure 4.** Ignition of samples of plywood (**a**), oriented stand board (**b**) and chipboard (**c**) at different wind speed. (I)—No Ignition zone; (II)—Ignition zone; Δ—Ignition; ×—No ignition.

Figure 5 shows a typical group of images on firebrand ignition of a chipboard sample, on the surface of which glowing firebrands of length 40 mm in the amount of six pieces were discharged.

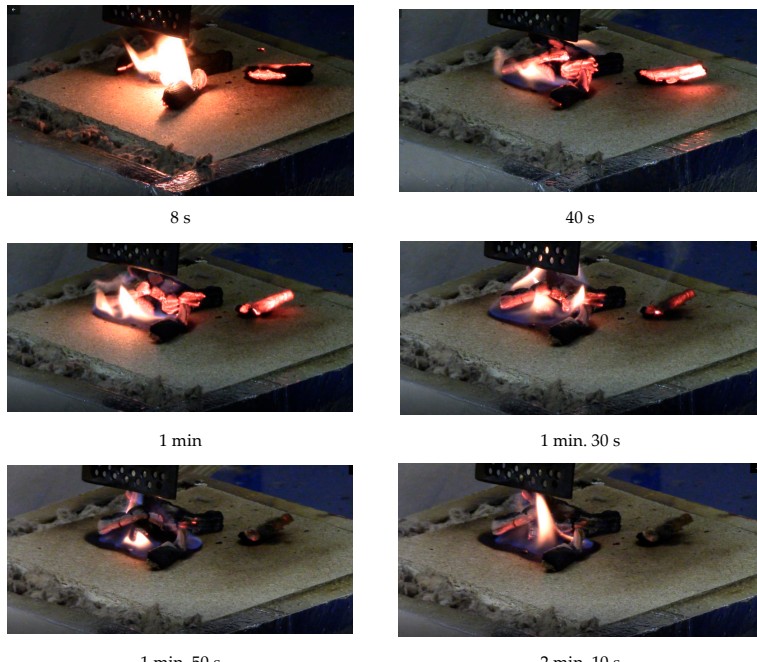

**Figure 5.** A group of images on the effect of firebrands on a chipboard sample. Figure 6 shows graphs comparing the ignition delay times of pine wood and construction materials (plywood and OSB-samples) depending on the number of firebrands at different wind speed.

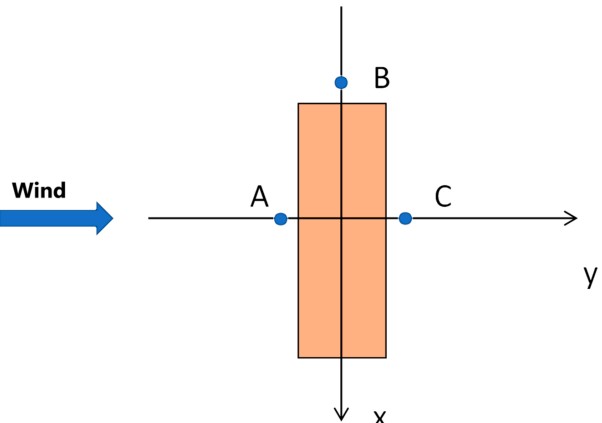

**Figure 6.** Layout of control points.

The wind speed was 2.0 m/s. It should be noted that the transition of firebrands from the glowing phase to the flame occurs due to the influx of the oxidizing agent from the heat blower. In particular, the transition occurred already at the 8th second in this case (Figure 5), which subsequently led to the burning of chipboard over the surface.

*3.2. Mathematical Modeling*

3.2.1. Single Firebrand Modeling

Let us proceed to the analysis of the results of mathematical modeling. The calculations simulated the ignition of a wood layer, both by a single firebrand and in the case of "fire rain", when several firebrands fall on the surface of the wood. Pine twigs were chosen as model firebrands.

Let us first consider the process of ignition of a layer of wood with a single firebrand. To study the process under consideration, the evolution of isotherms on the surface of the wood layer was analyzed, as well as the change in temperature over time at three characteristic points located at an insignificant distance (3 mm) from the windward (A), end (B), and leeward (C) edges of the firebrand. The arrangement of firebrands and control points is shown in Figure 6 (axes are selected in the direction of the wind).

The thermal energy stored in small firebrands is insufficient to initiate ignition. Therefore, the interaction of such firebrands with a wood surface occurs in a low-temperature regime (Figure 7). The adjacent layers of wood are heated (Figure 7a) as a result of heat transfer with the firebrand in the flow. The heating area increases in the initial period of time (Figure 7b). An increase in the heating area leads to firebrand cooling. As a result, the intensity of smoldering in the firebrand first decreases, and ultimately stops (Figure 7c). Subsequently, heat transfer with the environment leads to cooling of the firebrand and the adjacent layer of wood. In this case, the firebrand temperature approaches the ambient temperature (Figure 7d).

To initiate the ignition process, the smoldering firebrand must have high thermal energy. The calculations showed that a single firebrand with a size of L = 50 mm, H = 5 mm, D = 16 mm, and a temperature $T_{part} \leq 850$ K would not initiate the ignition of the wood layer. Since the maximum firebrand temperature is limited by the smoldering temperature, an increase in the thermal capacity of the firebrand is only possible with an increase in the firebrand size. Thus, only the thermal energy of large firebrands can provide the ignition.

Figure 8 shows isotherms on the surface of a wood layer in the vicinity of an igniting large firebrand at different moments. In this case, a high-temperature regime of heat transfer is implemented.

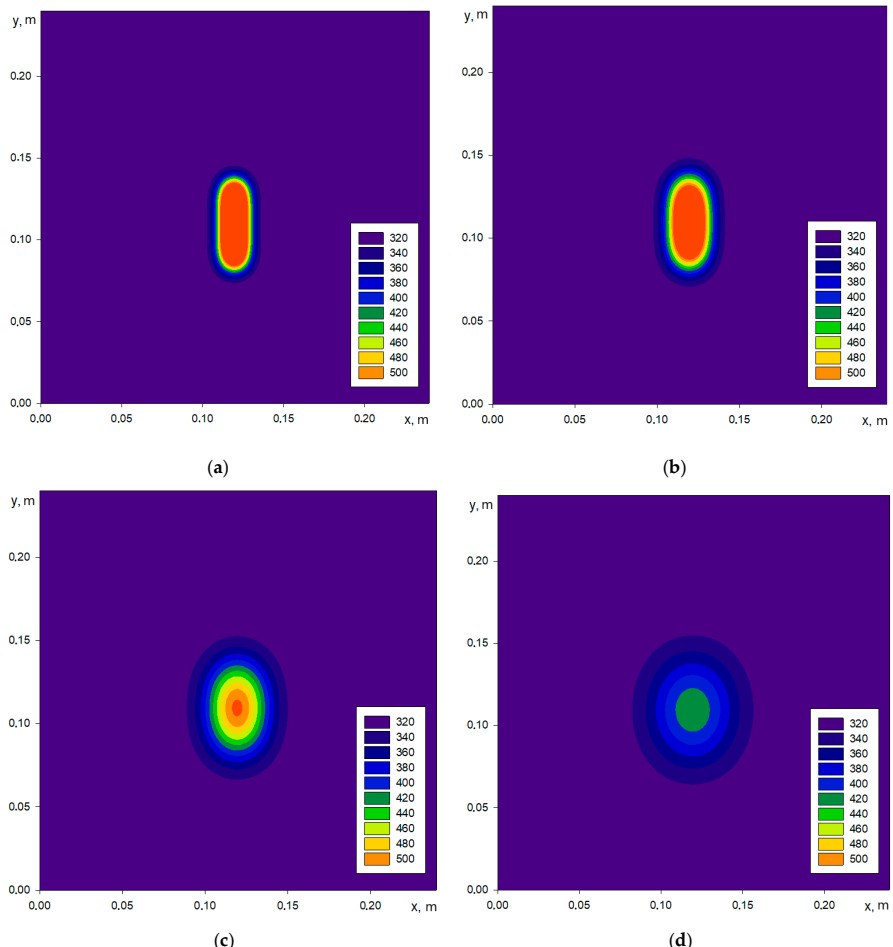

**Figure 7.** Isotherms on the surface of a wood layer in the vicinity of an igniting large firebrand
($L = 50$ mm; $H = 5$ mm; $D = 8$ mm) at different time: (**a**)—1 s; (**b**)—2 s; (**c**)—5 s; (**d**)—10 s; wind
speed—2 m/s.

In the high-temperature regime, as the temperature rises in the wood layer, the pyrolysis reaction is intensified. Gaseous pyrolysis products mix with atmospheric oxygen, the temperature in the channel wall reaches the adiabatic combustion temperature $T_a$, and a flash occurs. After the combustion of the pyrolysis products, the afterburning of the condensed products occurs in the smoldering mode. The process of interaction of a smoldering firebrand with a layer of wood goes through all the main stages in sequence: heating, drying, pyrolysis, ignition, and combustion. As can be seen from Figure 8, a high-temperature combustion zone is formed near the firebrand surface, which increases in size over time.

Figure 9 shows the temperature as a function of time for points A, B, and C, respectively. When performing the calculations, the length and height of the firebrands were assumed constant ($L = 50$ mm, height $H = 5$ mm), their thickness was varied in the range $D = 8 - 16$ mm.

Heat transfer of fine firebrands (curves 1, 2) is carried out in a low-temperature regime. As can be seen from Figure 9, the temperature of the wood layer at the points under consideration first increases, which is associated with the heating of the wood by the firebrand. Then, after the depletion of thermal energy reserves as a result of heat transfer with the environment, cooling occurs, and the temperature decreases.

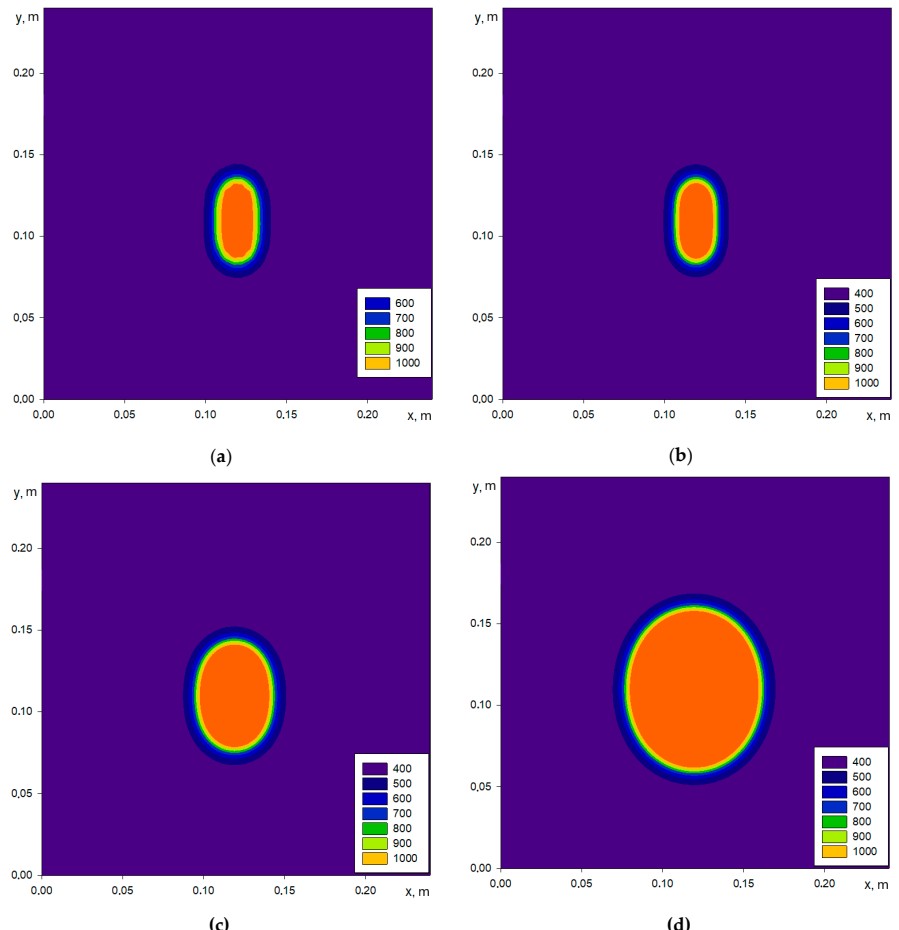

**Figure 8.** Isotherms on the surface of a wood layer in the vicinity of an igniting large firebrand ($L = 50$ mm, $H = 5$ mm, $D = 16$ mm) at different time: (**a**)—1 s; (**b**)—2 s; (**c**)—5 s; (**d**)—10 s; wind speed—2 m/s.

Curves 3–6 characterize the heat transfer of firebrands of large thickness and refer to the high-temperature regime. In the initial period of time, the temperature change over time is close to linear, which corresponds to the heating and drying of the wood layer. Then, after ignition of the wood layer, an exponential rise in temperature occurs. As the burnout progresses, the rate of temperature rise slows down. The temperature approaches the adiabatic combustion temperature, however, due to heat transfer, it does not reach its value. Then, as it burns out, the temperature decreases at the points under consideration. The cooling process after the burnout of the wood layer occurs at a sufficiently long time and is not shown in Figure 9.

The effect of the firebrand length on the heat transfer with the wood layer and the ignition process is illustrated in Figure 10. As can be seen, the low-temperature regime is observed for small firebrands. With increasing the firebrand length, the temperature of the wood layer increases. If the firebrand length exceeds the critical value, which depends on the size of other firebrands and heat transfer conditions, the ignition of the wood surface occurs. A further increase in the firebrand length leads to a decrease in the induction period and earlier ignition.

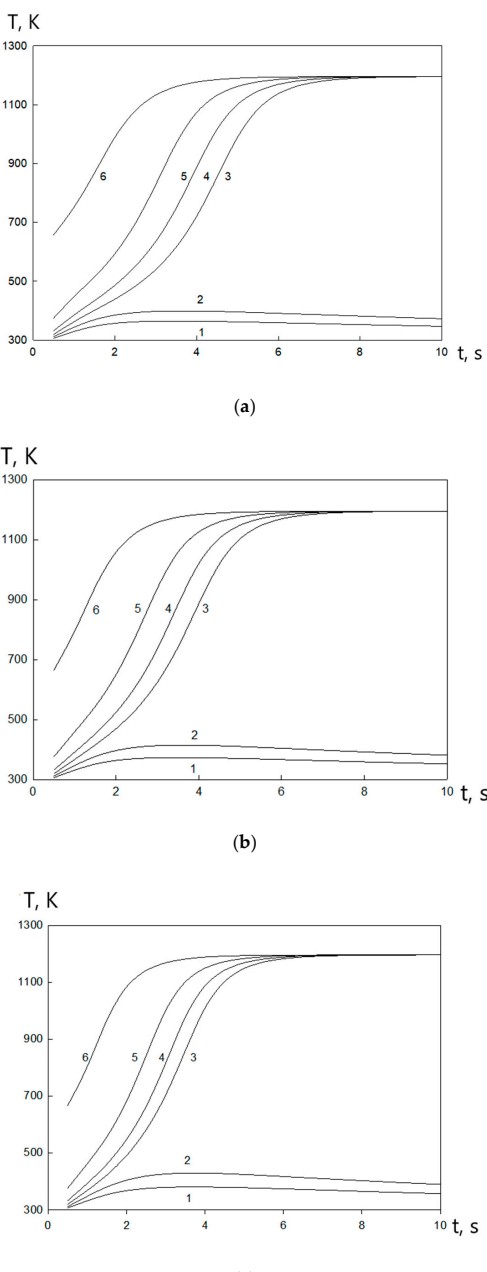

**Figure 9.** Temperature as a function of time: (**a**) point A, (**b**) point B, (**c**) point C, $L = 50$ mm, $H = 5$ mm, $1 - D = 6$ mm, $2 - D = 8$ mm, $3 - D = 10$ mm, $4 - D = 12$ mm, $5 - D = 14$ mm, $6 - D = 16$ mm.

The results of the study show that for the low-temperature regime, the temperature at the points under consideration differs insignificantly. As to the high-temperature regime, the heating and drying of the wood layer at all considered points proceed almost identically (this is indicated by the closeness of temperature curves at t = 0). However, the critical ignition conditions (the inflection point on the temperature curves) are reached first on the leeward side, then at the median point B, and only after that on the windward side.

Let us proceed to the analysis of the effect of firebrand size on the ignition conditions of the wood surface. Figure 11 shows the critical firebrand sizes, at which the ignition of the wood surface can occur.

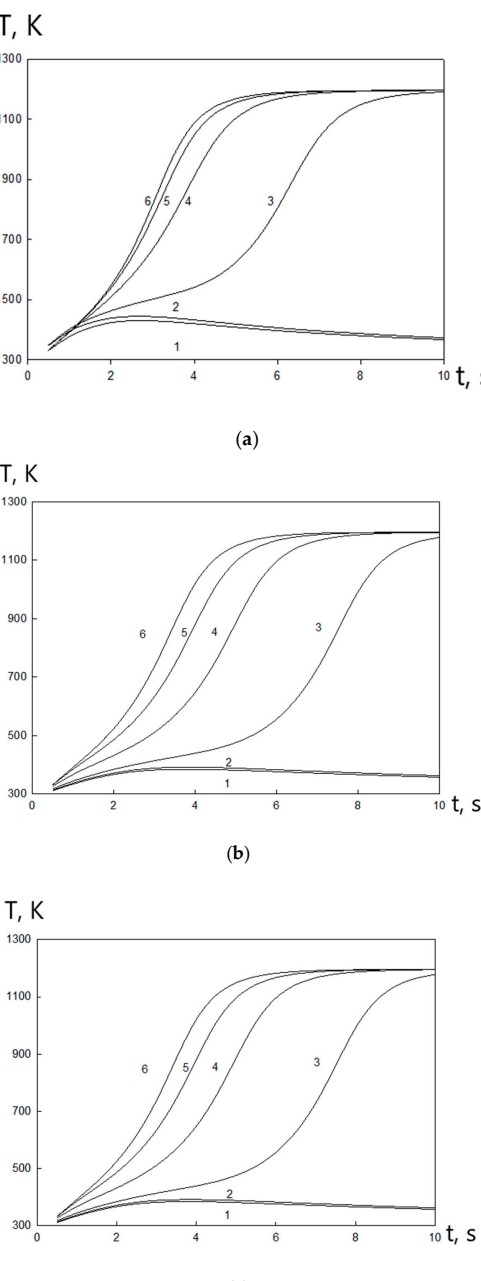

**Figure 10.** Temperature as a function of time: (**a**) point A, (**b**) point B, (**c**) point C, $D = 12$ mm, $H = 5$ mm, $1 - L = 16$ mm, $2 - L = 18$ mm, $3 - L = 20$ mm, $4 - L = 30$ mm, $5 - L = 40$ mm, $6 - L = 50$ mm.

Note that on Figure 11 the case $L < D$ corresponds to the orientation of the firebrand in the wind direction, for $L > D$ the firebrand is oriented perpendicular to the wind direction. The case $L = D$ corresponds to the firebrand with a horizontal square cross-section, so the orientations of the firebrand along and across the flow are the same.

The ignition of a wood layer requires a significant amount of thermal energy, which larger firebrands can possess. Thus, to initiate ignition, a decrease in the firebrand size must be accompanied by an increase in the other parameter.

Blowing the smoldering firebrand and wood layer leads to the intensification of heat transfer between them, and also contributes to the oxygen inflow to pyrolysis products, which intensifies combustion and leads to a decrease in the maximum size of the smoldering firebrand. On the other hand, intensive blowing leads to significant dilution and entrainment of pyrolysis products from the smoldering firebrand, as well as to a decrease

in the temperature and thermal energy of the firebrand. This effect was also observed in the experiments [73].

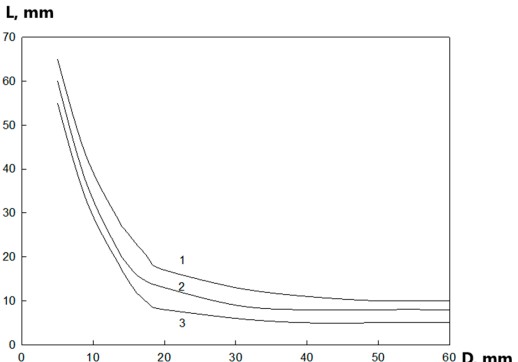

**Figure 11.** Critical dimensions of twigs providing ignition: $H = 5$ mm, $1 - v = 1.5$ m/s, $2 - v = 2.5$ m/s, $3 - v = 3.5$ m/s.

As a result of the difference in blowing the firebrands located along and across the wind flow, the dependencies shown in Figure 11 are not symmetrical to the straight line drawn at an angle of 45° to the axes. The critical thickness of a 60 mm twig located along the wind flow with a speed of 1.5 m/s is 8 mm and 5 mm for the same twig oriented across the wind flow. This fact is in agreement with the experimental data we observed.

Thus, the critical volume of the firebrand depends on the ratio of lengths of its sides and the orientation of the firebrand along the flow. The calculations show that the firebrands with a horizontal square cross-section have the minimum volume to initiate the surface ignition. For firebrands with a horizontal rectangular cross-section L≠D the critical volume increases the more the ratio L/D differs from unity. For firebrands oriented along the flow, the critical volume is higher than for firebrands oriented across the flow.

3.2.2. Modeling of the Interaction between a Group of Firebrands and a Wood Sample

The next step was the mathematical modeling of the interaction between a group of firebrands and a wood sample.

In mathematical modeling, it is assumed that the firebrands are located with their largest size perpendicular to the wind direction, symmetrically to the OX axis. In this case, the firebrands are parallel to each other and at the equal distance (Figure 12). A point located on the OX axis at the distance of 1 cm to the right from the edge of the rightmost (in the direction of the wind) firebrand is chosen as a control point (D).

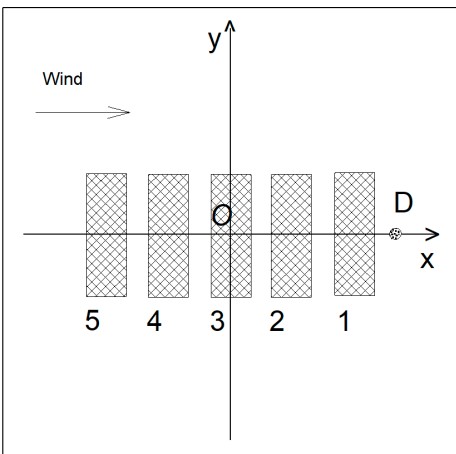

**Figure 12.** Layout of firebrands.

It is advisable to analyze the effect of the group of firebrands on the ignition process for such size firebrands, which single hit on the wood layer does not lead to ignition. Indeed, if a single firebrand is able to initiate the ignition process and combustion, then the hit of a group of such firebrands on a wood layer will cause its ignition.

Figure 13 shows the temperature distribution at point D caused by one thin twig hitting the wood layer ($L$ = 70 mm, $H$ = 5 mm, $D$ = 5 mm). As one can see from Figure 13, the layer heats up to about 380 K at the initial stage. However, the thermal energy stored in the firebrand is insufficient for further heating, drying, and initiation of a chemical reaction. Thermal energy decreases as a result of heat transfer with air and the wood layer, which leads to cooling and a decrease in temperature.

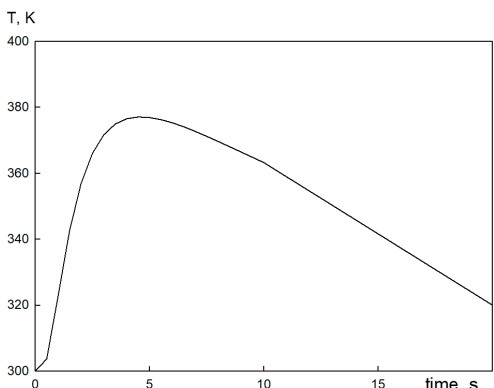

**Figure 13.** Temperature change at the control point caused by one thin twig hitting the wood layer over time: $L = 70$ mm, $H = 5$ mm, $D = 5$ mm.

Figure 14 shows the change in temperature at the control point (D) over time at different distances between the igniting firebrands.

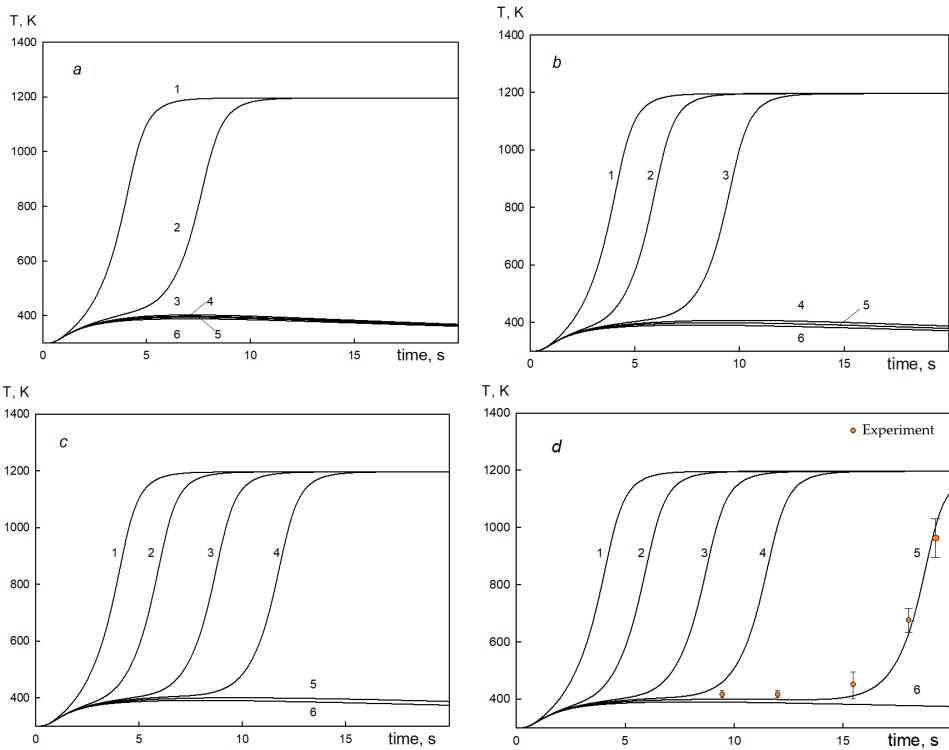

**Figure 14.** Temperature change at a control point over time: $L = 70$ mm, $H = 5$ mm, $D = 5$ mm, (**a**): 2 firebrands; (**b**): 3 firebrands; (**c**): 4 firebrands; (**d**): 5 firebrands ($1 - l = 0$, $2 - l = 15$ mm, $3 - l = 20$ mm, $4 - l = 25$ mm, $5 - l = 30$ mm, $6 - l = 35$ mm).

The results of mathematical modeling of heat transfer and wood ignition with two firebrands is considered next.

After the firebrands hit the wood layer near their boundaries, a temperature boundary layer is formed, which eventually spreads over the wood surface. Its vicinity begins to warm up after the temperature front reaches the control point (about 0.5 s).

Since a single firebrand of the considered sizes does not initiate the ignition of the wood layer, the high-temperature regime of heat transfer in the vicinity of the considered point is implemented as a result of the energy supply from both firebrands. In other words, it is necessary to reach the given point by a high-temperature front by the nearest as well as the distant firebrand for ignition.

The calculation results show that two thin twigs, which are located in close proximity to each other ($l = 0$), cause ignition of the wood layer (Figure 14a). At the same time, the processes of heating and drying proceed with high intensity, which leads to rapid ignition of wood.

If there is some distance between the firebrands, then part of their thermal energy will be spent on heating the gap between them. As a result, the rate of heating of the outer region slows down. However, if the firebrands are close to each other, the energy supplied from them may be sufficient to ignite. Thus, for closely spaced firebrands, their thermal interaction with the wood layer occurs in the ignition mode. The ignition time $t_*$ increases in a nonlinear manner with increasing distance between them and tends to infinity as $\rightarrow l_{*2}$. The distance between the firebrands $l_{*2}$ defines the boundary of the transition from the high-temperature regime to the low-temperature. It should be noted that for a practical assessment to determine the critical distance $l_{*2}$, the distance between the firebrands can be considered at which the ignition process does not occur for 1 min, and the cooling process occurs after the initial heating.

In the low-temperature regime (at $l \geq l_{*2}$) after a slight initial heating of the wood layer (after about 5 s), it cools down. The proximity of the temperature curves, describing the low-temperature regime, allowed one to conclude that the temperature profile evolution at the control point is determined by the effect of only the most closely located firebrand. The effect of the rest of the firebrands is negligible.

Figure 14b shows the temperature change at the control point for three smoldering firebrands. It can be seen that the main regularities in the evolution of the temperature distribution are preserved. Ignition occurs at small distances between firebrands $l < l_{*3}$. A low-temperature regime of heat transfer is implemented at large distances. The calculation results show that at small distances between firebrands $< 0.5l_{*2}$, the ignition process is determined by the heat supply from the two most closely spaced firebrands' side. Indeed, for $l < 0.5l_{*2}$ ignition occurs faster than the thermal front from the most distant firebrand reaches the considered point. The effect of the third firebrand starts at $0.5l_{*2} < l$. In addition, heat input promotes more intensive heating and a decrease in ignition time in comparison with two firebrands (compare curves 2 in Figure 14a,b). Furthermore, the ignition mode becomes possible at $l_{*2} < l < l_{*3}$ as a result of additional heating. If the firebrands are located far from each other ($l_{*3} < l$), then their thermal energy is insufficient for heating, drying, and pyrolysis of the wood layer. In this case, a low-temperature response regime is implemented.

The previously described regularities are also preserved for four firebrands (Figure 14c). The effect of the most distant firebrand is insignificant when the firebrands move away from each other at the distance $l < 0.5l_{*3}$. In this case, the ignition of the layer is specified by the effect of the three nearest firebrands, since the thermal front from the most distant firebrand has not yet reached the area under consideration. The energy of the nearest firebrands is sufficient to ignite the layer at $0.5l_{*3} < l_{*3}$, but the energy supply from the fourth firebrand contributes to reduction of the ignition time. In addition, the presence of the fourth firebrand ensures the ignition of the layer at $l_{*3} < l$. However, the high-temperature regime of heat transfer becomes impossible due to the limited reserves of thermal energy of firebrands at $l_{*4} < l$.

An increase in the number of firebrands to five makes it possible to carry out ignition at even greater distances between firebrands $l_{*4} < l < l_{*5}$ (Figure 14d). It should be noted that the ignition regime is characterized by a significant induction period at this range of distances between firebrands. The ignition time increases significantly (Figure 14d, curve 5), since the heat front from the nearest firebrands, as well as from the most distant, must reach the vicinity of the given point to ensure ignition. Comparison of the calculated and observed temperature change at a control point over time during experiment (in case of five firebrands) have shown the good agreement (Figure 14d).

It was found that ignition of the samples was not observed in the wind speed range of 0–1 m/s. A small amount of additional oxidant influx in the landing zone of firebrands resulted in the fact that the transition to intense flame combustion did not occur, and the smoldering of firebrands continued until their complete combustion. Thus, it can be concluded that smoldering wildland firebrands with the selected experimental parameters does not affect the ignition of samples of wood building materials. A similar report was observed in [28]. Ignition was found to be dependent on target fuel density; flaming ignition was additionally found to be dependent on wind speed. Higher wind speeds increased the rate of oxidation and led to higher temperatures and heat fluxes measured on the test surface [28].

## 4. Discussion

The results of investigations demonstrate that there is a tendency to decrease the ignition time of samples with an increase in the number of firebrands according to the analysis of graphs (Figure 15). It can be seen that in the cases with plywood, the ignition times are close to the times of pine wood ignition, and in the case with OSB, a large number of firebrands (8–10 firebrands) are observed, the ignition time is significantly reduced, more than two times compared to a similar experiment with pine. The obtained data allow one to judge that, at the chosen experimental parameters, the ignition time decreased with increasing wind speed, as well as with an increase in the number of firebrands.

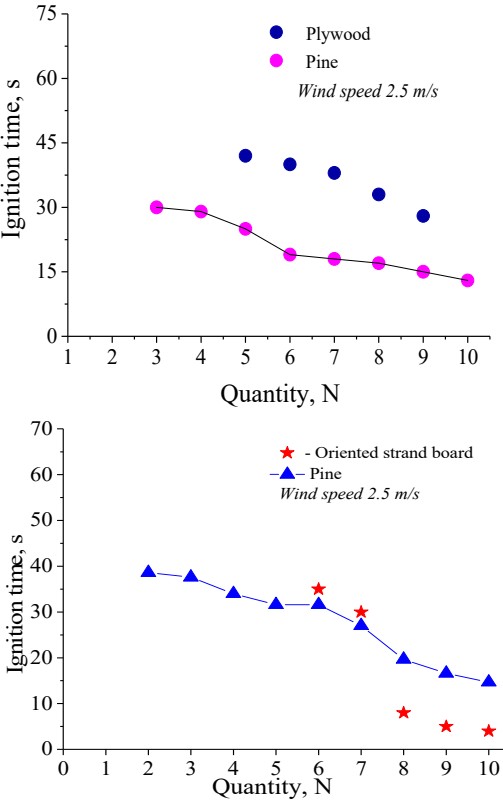

**Figure 15.** The ignition delay times of pine wood and construction materials.

The minimum total area of smoldering firebrands, equal to the product of the number of firebrands, sufficient for ignition, and their characteristic area, increases significantly during the transition from spruce wood [22] to wood building materials. In particular, for OSB and PB samples the minimum area increased by 40% on average, and for plywood sample the area increased by more than 60% at a speed of 2.0 m/s.

Figure 16 shows the critical distance between firebrands $l_{*N}$, separating the two heat transfer regimes, calculated for various numbers of firebrands.

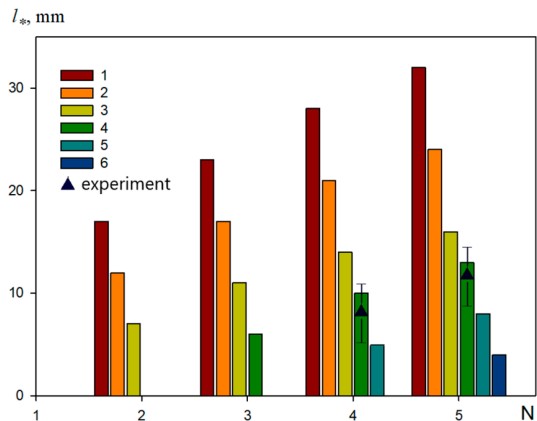

**Figure 16.** Critical distance between firebrands: $H = 5$ mm, $D = 5$ mm, $1 - L = 70$ mm, $2 - L = 60$ mm, $3 - L = 50$ mm, $4 - L = 40$ mm, $5 - L = 30$ mm, $6 - L = 20$ mm.

The figure shows that the ignition mode becomes possible at a higher distance between firebrands with an increase in the number of them. With a decrease in the firebrand length, the ignition of wood is possible with a decrease in the distance between the firebrands. In this case, ignition by small firebrands is possible only with an increase in their number. Thus, the ignition of wood with two firebrands is possible with their minimum length of 45 mm, three firebrands—35 mm, four firebrands—25 mm. Comparison of the calculated and observed critical distance between firebrands during experiment (for the case of $L = 40$ mm) have shown that our modeling accords well with the experimental results.

## 5. Conclusions

The paper presents a 3-D physical-based mathematical model of the ignition of a layer of wood by glowing wildland firebrand. A characteristic feature of the model is that it considers heat exchange between the firebrands, wood layer and the gas phase, moisture evaporation in the wood layer, pyrolysis and ignition of the firebrand and wood layer, and the diffusion and convection gases. In addition, the model takes into account the effect of the group of firebrands on the ignition process.

In order to test and verify the model, a series of experiments to determine behavior of wood construction material samples (plywood, oriented strand board, chipboard) in laboratory conditions as a result of exposure from pine twigs. The experiments tested the firebrand impact on the wood layer under different parameters, such as firebrand quantity, wind speed, type of wood material.

Irrespective of the pine twig quantity as used in the study, ignition of wood materials has not been observed in the wind speed range of 0–1 m/s. A small amount of additional oxidant influx in the landing zone of firebrands resulted in the fact that the transition to intense flame combustion did not occur, and the smoldering of firebrands continued until their complete combustion. It should be noted that the ignition time decreased with increasing wind speed, as well as with an increase in the number of firebrands. In particular, when the wind speed increases from 2 to 2.5 m/s, the minimum number of firebrands, sufficient to ignite wood, decreases from nine to four; eight to six; and six to four firebrands for plywood, OSB and chipboard, respectively.

The results of the mathematical modeling of wood ignition by pine twigs have shown that a single firebrand with a size of $L$ = 50 mm, $H$ = 5 mm, $D$ = 16 mm, and a temperature $T\_part \leq$ 850 K would not initiate the ignition of the wood layer. Flame combustion of the wood layer was observed only with firebrand sizes of $L$ > 7 cm.

Our model also demonstrated that the critical volume of the firebrand depends on the ratio of lengths of its sides and the orientation of the firebrand along the flow. For firebrands oriented along the flow, the critical volume is higher than for firebrands oriented across the flow.

Modeling of the interaction between a group of firebrands and a wood sample showed that in the low-temperature regime, the evolution of the temperature profile is determined by the effect of the closest firebrand only. With a decrease in the firebrand length, the ignition of wood is possible with a decrease in the distance between the firebrands. In this case, ignition by small firebrands is possible only with an increase in their number.

In order to model all aspects of the wood ignition by firebrands, it is necessary to develop the mathematical model describing accumulations of firebrands, which is the next stage of our research. Further work is needed to adapt and improve the physical and mathematical model proposed for describing the ignition the probability of ignition of certain types of structural materials (flooring, fence, as well as angular building structure) in the conditions of "firebrand shower".

**Author Contributions:** Conceptualization, D.K., O.M. and E.L.; methodology, D.K. and O.M.; validation, D.K, E.L. and A.L.; formal analysis, D.K. and O.M.; investigation, D.K., O.M. and O.D.; writing—original draft preparation, D.K., A.L. and O.M.; writing—review and editing, D.K., E.L. and O.M.; visualization, D.K., O.M., A.L. and O.D.; supervision, O.M.; project administration, D.K.; funding acquisition, D.K. All authors have read and agreed to the published version of the manuscript.

**Funding:** This research was funded by Russian Science Foundation, grant number 20-71-10068.

**Data Availability Statement:** The data presented in this study are available in the article.

**Conflicts of Interest:** The authors declare no conflict of interest.

## Nomenclature

| Term | Meaning (Units) |
| --- | --- |
| *Nomenclature* | |
| $C$ | heat capacity (J kg$^{-1}$ K$^{-1}$) |
| $\rho$ | density (kg m$^{-3}$) |
| $T$ | temperature (K) |
| $\lambda$ | thermal conductivity (W m$^{-1}$ k$^{-1}$) |
| $U$ | radiation intensity (W m$^{-2}$) |
| $\delta$ | the characteristic particle size (m) |
| $Q$ | reaction heat of pyrolysis, drying and combustion of coke (J kg$^{-1}$) |
| $\phi$ | the rate of pyrolysis, drying and combustion (coke) |
| $\varepsilon^*$ | is the degree of blackness of the wood layer |
| $\beta$ | integral radiation absorption coefficient ( m$^{-1}$) |
| $\chi$ | integral radiation attenuation coefficient ( m$^{-1}$) |
| $\sigma$ | is the black body radiation constant |
| $k$ | pre-exponential factor |
| $M_{hum}$ | relative moisture content of wood sample |
| $M_1$ | mass fraction of dry organic matter |
| $M_{pyr}$ | mass fraction of condensed pyrolysis products |
| $M$ | mass fraction |
| $\dot{h}$ | rate of the enthalpy change (J kg$^{-1}$ s$^{-1}$) |
| $E$ | activation energy, (J mole$^{-1}$) |
| $v$ | wind speed (m s$^{-1}$) |
| $g$ | gravity factor (m s$^{-2}$) |
| $\Gamma$ | coefficient of diffusion (m$^2$ s$^{-1}$) |

| | |
|---|---|
| $W$ | molecular mass, (kg mole$^{-1}$) |
| $p$ | pressure, Pa |
| $\mu$ | viscosity, (Pa s) |
| $R_g$ | universal gas constant (J mole$^{-1}$K$^{-1}$) |
| $C_{p,g}$ | heat capacity of gas at constant pressure, (Pa) |
| $t$ | time (s) |
| $x, y, z$ | coordinates (m) |
| *Subscripts* | |
| $fb$ | firebrand |
| $f$ | solid phase (wood, firebrand) |
| $td$ | thermal degradation |
| $pyr$ | pyrolysis |
| $dr$ | drying |
| $w$ | wood layer |
| $g$ | gas |
| 1 | dry organic substance |
| 2 | water in the liquid-drop condition |
| 3 | condensed pyrolysis products |
| $O_2$ | oxygen |
| $CO$ | carbon monoxide |
| $CO_2$ | carbon dioxide |
| $N_2$ | nitrogen |
| $H_2O$ | water |
| O | ambient |

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
