# Peer review of "Modeling of Wood Surface Ignition by Wildland Firebrands"

_fire, doi:10.3390/fire5020038_

Round 1
Reviewer 1 Report
All the reviewer's comments have been taken into account and the article can be published.
Author Response
Thank you for your kind comments.
Reviewer 2 Report
This is a re-submitted article, and authors have tried to modify their article to some degree based on reviewers comments, and consequently, the quality of this article is now better than before. Unfortunately, the authors have technically ignored many important recommendations, which are still necessary to be addressed in this article for the passing of the acceptance criterion of mine. I am going to include here one of my recommendations for your convenience.
=================================================
10. Finally, the conclusion part MUST also be precise and straightforward as an abstract so that the potential readers can easily understand events as mentioned above (1) along with major findings of the article. Again, the ‘conclusion’ part is not written as expected for a scientific article. Besides, the ‘conclusion’ must have consistency with the abstract; this is a common practice of writing a reader-friendly scientific article. Hence, a major revision is necessary as suggested above.
=================================================
In the current version, the ''Discussion and Conclusions'' have been compiled in one section, which is the direct ignorance of my recommendation 10, as mentioned above. Therefore, I am recommending authors to check my previous comments point by point with special attention.
Author Response
Thank you for your valuable comments and suggestions. I have presented below answers to questions:
(Point 1): This is a re-submitted article, and authors have tried to modify their article to some degree based on reviewers comments, and consequently, the quality of this article is now better than before.
Response 1: Thank you very much for your positive comments.
(Point 2): Unfortunately, the authors have technically ignored many important recommendations, which are still necessary to be addressed in this article for the passing of the acceptance criterion of mine. I am going to include here one of my recommendations for your convenience.
=================================================
10. Finally, the conclusion part MUST also be precise and straightforward as an abstract so that the potential readers can easily understand events as mentioned above (1) along with major findings of the article. Again, the ‘conclusion’ part is not written as expected for a scientific article. Besides, the ‘conclusion’ must have consistency with the abstract; this is a common practice of writing a reader-friendly scientific article. Hence, a major revision is necessary as suggested above.
=================================================
In the current version, the ''Discussion and Conclusions'' have been compiled in one section, which is the direct ignorance of my recommendation 10, as mentioned above. Therefore, I am recommending authors to check my previous comments point by point with special attention.
Response 2: The authors are grateful for the essential comments. During the revision the abstract section was fully revised. The Introduction section was also revised and broadened. Based on an extensive literature review, the focus of this research, as well as a characteristic feature of the model presented with its subsequent verification has been showed. A list of nomenclature was added to the paper as well as the comparison with the experimental results. According to reviewer’s comments we added to the paper heading 2.2 - "Method of Solving the Problem".
We agree with reviewer’s remark about the section "Discussion and Conclusions". The authors organized a separate section "Discussion" , in which we presented the relevance of results regarding the findings of the model and the experiment, comparing them with each other. The conclusion section was revised to show the major findings of the article.
Reviewer 3 Report
The present manuscript is an important contribution to the understanding of wood surface ignition by wildland firebrands.
I believe that the authors were able to answer the questions of previous reviewers, resulting in the improvement of scientific writing, facilitating the understanding of fire readers. However, I would like the authors to check a few points before final editing for publication.
Line 26: “0÷1 m/s”, the correct one would be: “0–1 m/s”.
Line 143: “[]” What is the scientific reference?
Author Response
Thank you for your valuable comments and suggestions. I have presented below answers to questions:
(Point 1): The present manuscript is an important contribution to the understanding of wood surface ignition by wildland firebrands.
I believe that the authors were able to answer the questions of previous reviewers, resulting in the improvement of scientific writing, facilitating the understanding of fire readers.
(Response 1): Thank you for your positive evaluation of our work.
(Point 2): Line 26: “0÷1 m/s”, the correct one would be: “0–1 m/s”.
Response 2: We agree with reviewer’s remark. The sentence has been corrected.
(Point 3): “[]” What is the scientific reference?
Response 3: There was a technical error, the reference was at the beginning of the sentence. "[]" has been deleted from the sentence.
Round 2
Reviewer 2 Report
I am writing my comments very briefly bellow:
(a) The revised abstract is not yet self-dependent summary of the whole work. Unnecessary words/phrases and sentences have been inserted in the abstract, which may be suitable elsewhere in this article. This remains as teh one of the major drawbacks of this article.
(b) Another shortcoming of this article is to the representation of 'research gap' systematically, which is still not clear. I believe that no potential readers will be able to figure it out easily. Without revealing 'research gap' properly and systematically, any research has no scientific value as the 'research gap' basement over which all of the article has been constructed so as to fill the ‘research gap’.
(c) In section, 2.1. Mathematical mode has been presented and later on it has been stated that mathematical equations has been solved by numerically using the finite-volume method. Well; but no information available whether authors have applied in-house code or commercial codes!!. Moreover, the major numerical results should be presented just after the numerical section and before experimental section. The potential reader will encounter very difficulty in finding the results of this section. This is another drawback of this article!
(d) The potential reader would like see the experimental results against the figure 7 and 8 along with their comparison, which is missing in this version.
(e) The ‘Discussion’ section is seen in 576. So, naturally potential readers will rise a question: What does it mean the literature with Figs that is presented before this ‘Discussion’ section that is found at the line 576??
(f) I am very sorry to say again the same thing that the conclusion part is NOT self-dependent and straight forward along with the point wise representation of major findings. This is really a crucial event and shortcomings as well for this article!!
Anyway, please wait for the comments from the editorial office.
Author Response
Thank you for your valuable comments and suggestions. I have presented below answers to questions:
(Point a): The revised abstract is not yet self-dependent summary of the whole work. Unnecessary words/phrases and sentences have been inserted in the abstract, which may be suitable elsewhere in this article. This remains as teh one of the major drawbacks of this article.
Response a: The abstract has been corrected.
(Point b): Another shortcoming of this article is to the representation of 'research gap' systematically, which is still not clear. I believe that no potential readers will be able to figure it out easily. Without revealing 'research gap' properly and systematically, any research has no scientific value as the 'research gap' basement over which all of the article has been constructed so as to fill the ‘research gap’.
Response b: We agree with reviewer’s remark. The introduction section has been revised. After a comprehensive literature review, the authors problematize the question as to what only a few models take into account the contribution of burning and smoldering firebrands generated in the combustion zone, which are one of the main causes of fire spreading around the world. The presence theories have multiple limitations because it mostly does not take into account ongoing reactions in firebrands, the moisture content of fuels, lack of good conductive contact between fuels and firebrands, radiative feedback or external radiation and thermophysical properties of the fuel.
In this regard, the authors propose to a 3-D physical-based mathematical model of wood surface ignition by wildland firebrands that considers heat exchange between the wood layer and the gas phase, moisture evaporation in the firebrands and the diffusion of water vapour in the pyrolysis zone.
(Point c): In section, 2.1. Mathematical mode has been presented and later on it has been stated that mathematical equations has been solved by numerically using the finite-volume method. Well; but no information available whether authors have applied in-house code or commercial codes!!. Moreover, the major numerical results should be presented just after the numerical section and before experimental section. The potential reader will encounter very difficulty in finding the results of this section. This is another drawback of this article!
Response c: This is a valid point. We have applied own code developed by authors. This code has repeatedly tested and verified on a wide set of tasks. The results of verification of this computational code have showed its reliability and possibility to use it to solve the problem.
(Point d): The potential reader would like see the experimental results against the figure 7 and 8 along with their comparison, which is missing in this version.
Response d: The authors believe that it would be difficult for the potential reader to analyze the experimental results on the graphs with isotherms on the surface of a wood layer (Figures 7, 8). Experimental results are added on Figure 14, d.
This manuscript is a resubmission of an earlier submission. The following is a list of the peer review reports and author responses from that submission.
Round 1
Reviewer 1 Report
The article is very well written. Great literature review on the topic, references to previous studies and convergence of results.
There are a few comments to the article.
1. Abstract. There are no descriptions of the specific achievements cited in the article.
2. The objectives of the study are not defined. The results of solved tasks should be reflected in the conclusions.
3. line 231 - give the source of the definition in inverted commas.
4. Line 237-238 - unnecessary subdivision information. This is obvious.
5. Lines 246, 250 - wind speed - 2.0 (not 2).
6. Figure 3. Reduce from 8 to 3 cases? Specify for which type of material.
7. Line 330 - what was the ambient temperature during the experiments?
Reviewer 2 Report
Comments:
The authors have investigated the ‘Modeling of wood surface ignition by wildland firebrands’, which is somehow rich to some extent, but it is not well-organized. However, this article still suffers from significant drawbacks and some of them are listed below:
- Usually, the abstract of a reader-friendly, scientific article is the ‘self-dependent and concise summary’ of the whole investigation; this is a MANDATORY criterion. Unfortunately, the current version of the abstract does not satisfy the criterion, as stated above. So, a reader-friendly abstract MUST be written in such a way so that the potential readers can easily find the following information systematically for example ‘the definition of the problem of investigation’, ‘the precise description of the applied methodologies, and ‘the key findings of the investigation (along with future direction of this work, if any), which are unique and universally valid under the wide range of pertinent conditions. This is one of the major shortcomings of this article, and hence a major revision MUST be done in such a way so that the precise ‘research gap’ can be found easily by the potential readers. Besides, the current version of the abstract, unnecessary words/sentences are included; most of them are applicable in the introduction part.!! The authors are strongly suggesting eliminating it in the revised version.
- The next important and most vital part of an article is the ‘introduction’. It is generally treated as the heart of an article. The introduction part usually guides the flow of the rest of the article’s parts. So, a question naturally arises on how to construct the introduction? It is a significant event for a good and well-organized scientific article. To address the above question, it is required to survey the existing literature on the subject matter of the article extensively, and this kind of literature survey will help authors to reveal a ‘research gap or originality’ within the existing literature. Once the ‘research gap’ is identified, then the rest of the article MUST be devoted to filling up the ‘research gap’ as identified. This type of construction of an article as mentioned above usually enhances the understandability of the potential readers, even those who are not experts on the topic of investigation. Unfortunately, the ‘introduction’ of this article is not written as highlighted above. In other words, the authors have completely failed to reveal the ‘research gap’ in this investigation. Without revealing the ‘research gap’ systematically and precisely, strictly speaking, any kind of research work has no scientific value! This is another severe shortcoming of this article!
- The figures of this article are irregular in size (i.e., the figures should be uniform in size within the article, and the texts within the figures are too small as compared with the running texts. Hence the modification of the figures is recommended.
- A lot of equations have been included in the modelling sections and a list of nomenclature is mandatory for this article. Otherwise, just after the each or a set of equations, an explicit explanation is necessary for each term and coefficients involved in the equation, even though the equations are well-known ( for example, Navier Stokes equations!)
- The alignment between running texts and equations is inconsistent, and it is another obligatory event which requires modification for this article.
- The numerical simulation is conducted along with the experiment, but no comparison between them is found in this article and this one of the drawbacks of this article. So, this point MUST be addressed precisely.
- In numerical simulation, the usual norm is to write about the grid dependency test, whether the in-house code or commercial code is applied to carry out the numerical simulation. This event is absent in this article. Hence, the authors are suggested to address this event precisely from the revised article, and this is a MADATORY event.
- The boundary and initial conditions MUST be written in a systematic way in the modelling section, which is absent, and a revision is necessary in this regard.
- In this article, a huge error related to ‘gap’ is found. For example, in line 364, ‘L=D’ corresponds to the firebrand with a horizontal square cross-section. Hence, these types of errors MUST be eliminated in the revised throughout thearticle.
- Finally, the conclusion part MUST also be precise and straightforward as an abstract so that the potential readers can easily understand events as mentioned above (1) along with major findings of the article. Again, the ‘conclusion’ part is not written as expected for a scientific article. Besides, the ‘conclusion’ must have consistency with the abstract; this is a common practice of writing a reader-friendly scientific article. Hence, a major revision is necessary as suggested above.
- This article has many other minor errors, and hence the authors are suggested to check their article thoroughly to figure out them, and finally modify them accordingly.
- Anyway, please wait for the comments from the editor.